# Assessment of Operational Effectiveness of Innovative Circuit for Production of Crushed Regular Aggregates in Particle Size Fraction 8–16 mm

Tomasz Gawenda, Agata Stempkowska *, Daniel Saramak, Dariusz Foszcz, Aldona Krawczykowska and Agnieszka Surowiak

Department of Environmental Engineering, Faculty of Civil Engineering and Resource Management, AGH University of Science and Technology, Mickiewicza 30 Av., 30-059 Cracow, Poland; gawenda@agh.edu.pl (T.G.); dsaramak@agh.edu.pl (D.S.); foszcz@agh.edu.pl (D.F.); aldona.krawczykowska@agh.edu.pl (A.K.); asur@agh.edu.pl (A.S.)
* Correspondence: stemp@agh.edu.pl

**Abstract:** The purpose of this paper is to analyze a modern and unique technological system producing common aggregates at the Imielin Dolomite Mine. The installation was built on the basis of inventions of AGH UST and consists of an impact crusher, innovative screens WSR and WSL, light fraction separator SEL and hard fraction separator SET, low-pressure hydrocyclone NHC and infrastructure. The study was carried out on the crusher and screen on the example of production of aggregates with grain size 8–16 mm from dolomite, granite, limestone, sandstone, and gravel. The results showed that cubic aggregates with a low content of irregular grains of less than 1% can be produced in this technological system.

**Keywords:** regular aggregate; innovative installation; separation

## 1. Introduction

Requirements for aggregate quality and minimization of energy consumption pose increasing challenges for raw material processing plants. A demanding market may in future require large quantities of products with narrow grain size distribution and specific grain shapes. The physical and chemical properties of the material, such as density, hardness, strength, and structure, depend on the place of exploitation (the origin of the raw material), and therefore, are generally invariable in the processing operations in mineral processing. On the other hand, specific particle size and shape or surface structure are achievable depending on the processing methods used during their manufacture (especially crushing) and can directly affect other properties [1–3]. The most important rock raw materials for civil engineering are fractured aggregates produced from magmatic, sedimentary, and metamorphic rocks [4–6]. The protection of the earth's natural resources and the increased use of industrial wastes have resulted in parallel research on the use of recycled and artificial aggregates [7]. Aggregates of the highest quality (e.g., basalt, melaphyre) are used in various branches of construction. The created precast elements that carry enormous dynamic loads, are subject to direct abrasion and adverse weather conditions. These structures should be made of aggregates with low abrasiveness, high strength, and resistance to water and frost. In addition, these aggregates should be characterized by particle shape close to spherical or cubic, sharp edges and rough fracture surfaces. All types of aggregates are commonly used for road pavement layers and structural elements of all types of concrete [8–12].

In the ongoing scientific research on the issues of mineral aggregate production [13,14], it has been noted that the hardness of the raw rock affects its shape. In general, the more compact the raw material, the more difficult it is to obtain cubic particles from it [15–18].

Moreover, in the finer fractions of the crushing products, the largest number of elongated and flat grains is obtained. Therefore, it is advisable to use impact crushers (e.g., with a vertical shaft) at the final stages of crushing. As the crushing rate increases, the percentage of irregular (especially in fine fractions) aggregates also increases, thus crushers should be operated at a crushing rate that is not too high. However, all of these important factors influence the necessity to expand and use multi-stage systems, which usually lead to increasing both the investment and operating costs. There are many studies on the influence of grain geometry on the parameters (such as grinding ability) and properties of aggregates and materials made from them [19–22]. However, there is no data on how to produce cubic aggregates with a proportion of up to 100% apart from publications and patents by the author Gawenda et al. [14,23–26], and a few publications on how to increase the proportion of regular grains [13,27,28], especially in the industrial utilization of aggregate products in building industry [29–31].

*Full-Scale Prototype of an Innovative Technological Circuit*

In order to increase the quality of aggregates and reduce the number of crushing stages, research was undertaken as part of the implementation of the NCBiR application project: Action 4.1 of the Operational Programme Intelligent Development 2014–2020. HTS Gliwice in cooperation with scientific and research units (AGH in Krakow, ICIMB Łukasiewicz Research Network) built a full-scale prototype of an innovative technological system for refining mineral aggregates. The tests were carried out in real conditions. The installation of the technological system was placed in the Imielin Dolomite Mine and consisted of crushing, enriching, and classifying machines, as shown in Figure 1.

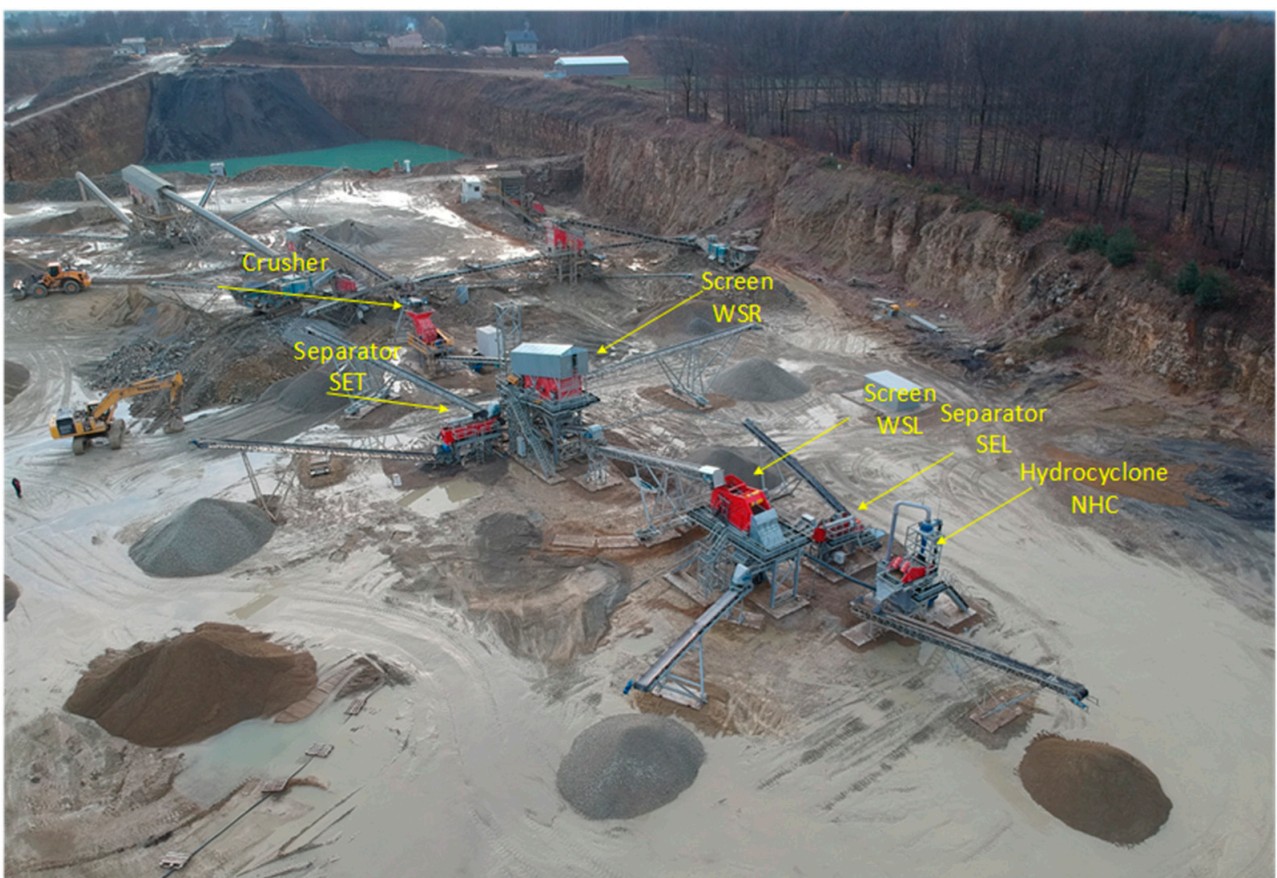

**Figure 1.** Technological line in the Imielin Dolomite Mine, image from a drone.

The process line includes:

I. *Formator*, which is an installation for the production of regular and irregular fractions consisting of: Impact crusher, three-deck rotary specialized vibrating screen (WSR), three-deck linear specialized vibrating screen (WSL), infrastructure (feeding and buffer cages, control and power supply, belt conveyors, chutes, pumps);

II. Light contaminant separator (SEL);

III. Separator of difficult-to-separate fractions (SET);

IV. Low pressure hydrocyclone (NHC).

A schematic of the pilot plant used for process testing on an industrial scale is shown in Figure 2. This is the unique plant in the world that includes the production of innovative refined aggregates with regular grains in the range of 2–8 mm and 8–16 mm and a sand fraction of 0.1–2 mm.

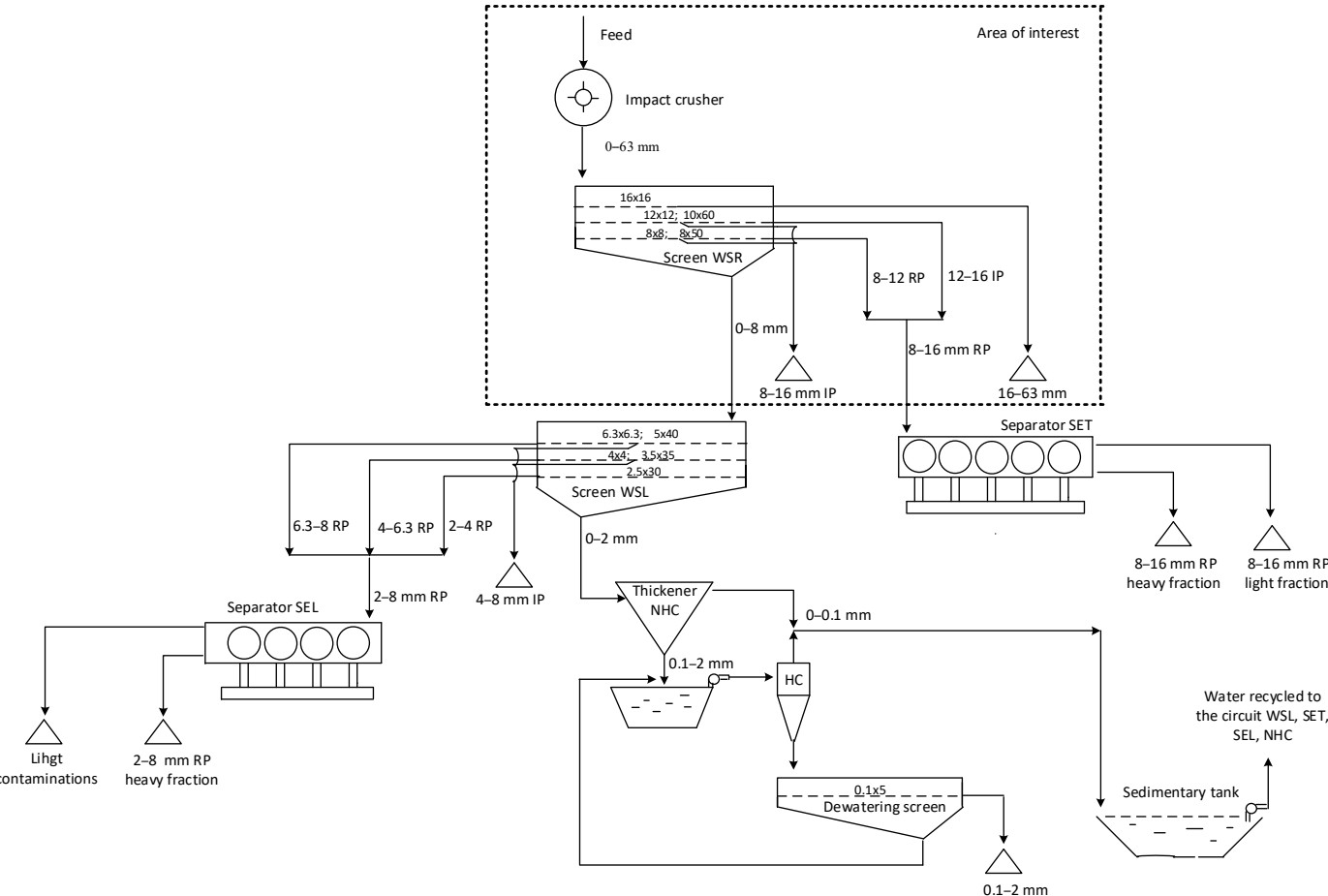

**Figure 2.** A simplified flow diagram of the installation for the mineral aggregates refinement in the Imielin Dolomite Mine.

The aim of this paper is to present the operation of the *Formator* technological system that produces 8–16 mm class of shaped aggregates (regular particles = RP) from various rock materials. The analyzed system consisting of an impact crusher and a WSR screen is marked on the technological diagram with a dashed line (Figure 2). It was built in accordance with the concept of the patent PL-231748B1. The assumption for the construction of this system was the possibility of producing aggregates with a content of non-formed grains (irregular particles = IP) below 3%, which is not possible in conventional technological systems.

Table 1 provides a brief comparison of an innovative device with a typical device.

**Table 1.** Advantages and disadvantages of the innovative and ordinary screens in the technological system for the production of regular aggregates.

| Innovative Screen Machine | Typical Screen Machine |
|---|---|
| Possibility of separation of regular aggregates with a low content of irregular grains <3%. | Lack of possibility of separation of regular aggregates. To achieve this effect, it would be necessary to build two screens sited one behind the other and connected by chutes or conveyor belts, which generates investment costs and the need for land area for development (Gawenda T. 2019: Equipment layout for production of molded aggregates Patent No. PL233689 B1, AGH in Krakow). The first screen separates narrow grain classes and the second screen on slot screens separates aggregates by shape. |
| The final (over-screen) products have a low content of sub-grain up to 2%. | Sub-grain content in products up to more than 10%. |
| The use of a screen in the crushing system eliminates one crushing stage in the production of cubic aggregates due to the screening of irregular grains, which can be re-grinded. | Need to divert all of the material to regrind to improve aggregate cubicity. |
| The need to sow narrow fractions to separate them into regular and irregular grains, which increases the number of beds, but then there is the possibility of separation of the narrow fractions, which stabilizes the uniform composition of the aggregate mix. | The fractions are not separated into regular and irregular grains, but to separate more narrow fractions the number of decks also needs to be increased. |
| Need for larger screening area (30% larger screen) or reduced capacity. | Higher efficiency. |
| Need to provide higher toss ratio (min. of 6 g) to eliminate clogging of the screen mesh. | Lower toss ratio. |

## 2. Materials and Methods

Characteristics of the technological unit-*Formator*.

### 2.1. Impact Crusher

The KU 80/120 impact crusher used in the technological system (Figure 3) was selected due to its advantages related to obtaining products with a lower content of irregular grains and higher strength of aggregates in comparison with products obtained from cone and jaw crushers [27]. Crushing occurs mainly due to the impact of material grains by the rotating slats attached to the rotor and the impact of the material accelerated by the slats on the stationary breaker plates. The adjustable gap between the rotor and plates allows for the adjustment of the size of the obtained fractions to the needs of the user. Properly selected materials, of which the slats and stationary plates are assembled, ensure their high resistance to abrasion, thanks to the low operating costs that the user incurs. Table 2 presents the characteristics of the crusher.

**Table 2.** Characteristics of the KU 80/120 crusher.

| Name | Unit | Value |
|---|---|---|
| Inlet Size | [mm] | 800 × 1170 |
| Rotor diameter | [mm] | 1110 |
| Rotor length | [mm] | 1150 |
| Capacity (maximum) | [t/h] | 450 |
| Feed grain size range | [mm] | 0–700 |
| Hard feed grain size | [mm] | 0–400 |
| Crusher weight | [kg] | 11,851 |
| Rotor weight | [kg] | 4136 |
| Crusher drive power (max.) | [kW] | 250 |
| Rotor Speed I | [rpm] | 529 |
| Rotor Speed II | [rpm] | 670 |

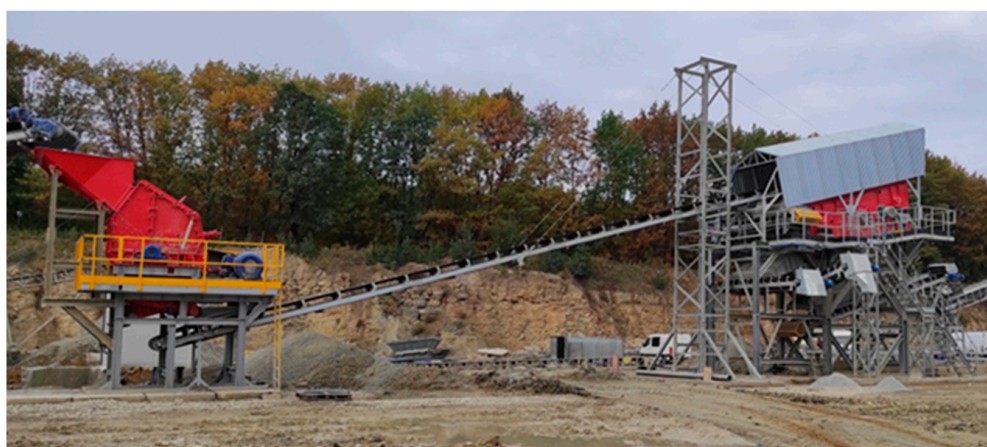

**Figure 3.** View of the KU 80/120 crusher and the WSR 3-2.0/6.0 screen.

### 2.2. Specialized Rotary Vibrating Screen (WSR)

Specialized rotary vibrating screen (WSR 3-2.0/6.0) (Figure 3) is a machine designed for screening the excavated material with transversely tensioned screens. The aggregate delivered to the screen deck is subjected to circular harmonic vibrations, where the material is periodically lifted under the force of gravity and moves along the screen deck. The elements causing the circular motion vibrations are rotating eccentric (shaft and inertial) masses. The vibrating force is directed at an angle of 15° to the base of the screen. The structure of the screen is shown in a schematic diagram in Figure 4. The technical characteristics are provided in Table 3. The screen is built in accordance with the concept of invention patented by AGH. It is an atypical screen (specialized) since it has three decks and produces six fractions. Its characteristic feature is first screening the feed material into fractions of 8–12 and 12–16 mm with the use of two decks equipped with square mesh screens, and then separating the irregular grains from the regular ones with the use of slot screens (rectangular mesh) (Figure 2). In the oversize-screen products, form aggregates are obtained, which, when joined together, form the innovative 8–16 mm RP (regular particles) products. In the undersize-screen products of these deposits, products with an increased proportion of 8–16 mm IP irregular particles are obtained after merging. Moreover, this screen separates the 0–8 mm fraction, which is fed to another specialized WSL screen, also to obtain the shaped aggregates. The fraction above 16 mm is a typical commercial product.

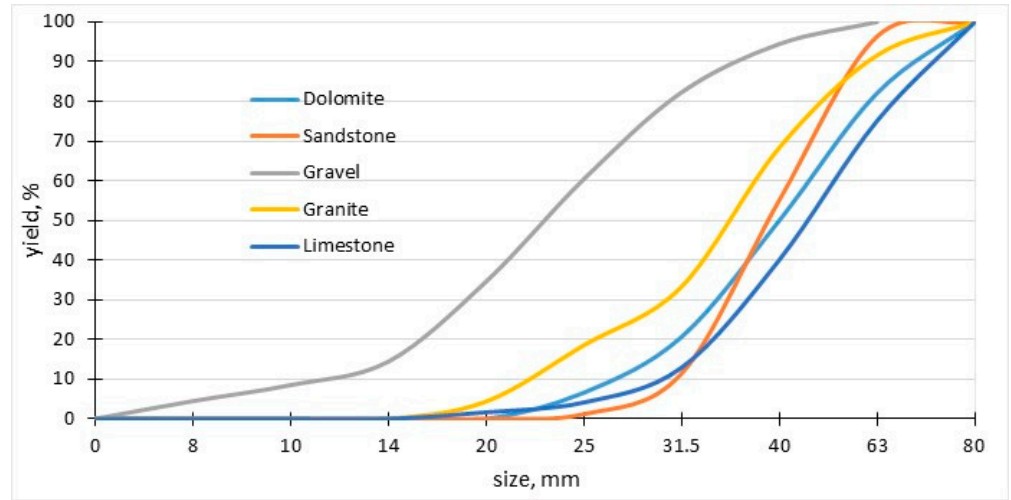

**Figure 4.** Particle size distribution curves of five feeds sent to the impact crusher.

**Table 3.** Characteristics of the WSR 3-2.0/6.0 rotary screen.

| Name | Unit | Value |
|---|---|---|
| Sieve dimensions | [mm] | 6000 × 2000 |
| Working frequency | [Hz] | 16.6 |
| Maximum stroke | [mm] | 9 |
| Angle of screen inclination | [°] | 15 |
| Drive power of screening unit | [kW] | 30 |
| Motor rotation | [rpm] | 980 |
| Maximum efficiency | [t/h] | 350 |
| Screen deck 1 | [mm] | half deck-16 × 16 mesh, half a deck-blank screen |
| Screen deck 2 | [mm] | half deck-12 × 12 mesh half deck-10 × (60) 75 gap |
| Screen deck 3 | [mm] | half a deck-8 × 8 mesh half deck-8 × (50) 60 gap |

## 3. Results and Discussion

### 3.1. Tests on Different Types of Raw Materials

The raw materials selected for testing were various in terms of lithology and grain size distribution. Triassic dolomite, Triassic limestone, sandstone from the Carpathian flysch, gravel of river origin, and granite from the Central Sudety region were used. The graph (Figure 4) presents the particle size distribution curves of the feeds sent for crushing in the impact crusher.

Analyzing the grain size composition of the feed material (Table 4), it should be emphasized that the finest grain size range is the river gravel 0–63 mm and the largest share of coarse grains by limestone is in the range 14–80 mm. The content of out-of-form grains measured with a Schultz caliper in accordance with PN-EN 933-4:2008 [32] (SI shape index) in the feed varied and ranged from about 18% for dolomite to 25% for sandstone.

**Table 4.** Characteristics of raw materials and operation of the crusher.

| Raw Material | Feed Grain Size [mm] | Shape Index SI [%] | | Comminution Degree | | Crusher Throughput [t/h] |
|---|---|---|---|---|---|---|
| | | Feed | Crusher Product | S90 | S50 | |
| dolomite | 20–80 | 17.8 | 14.9 | 2.8 | 6.7 | 80 |
| sandstone | 25–80 | 25.3 | 13.5 | 2.8 | 4.2 | 80 |
| gravel | 0–63 | 24.3 | 13.6 | 1.8 | 3.3 | 60 |
| granite | 14–80 | 20.0 | 15.7 | 2.2 | 3.9 | 100 |
| limestone | 14–80 | 18.6 | 13.7 | 3.4 | 6.0 | 100 |

The raw material was crushed using an impact crusher at a rotation speed of 529/min. The outlet gaps between the strips and the impact plates were 20, 40, and 60 mm. The finest grades (S50) obtained were highest for dolomite and limestone (about 6) and lowest for gravel, which is related to the grain size of the feed. The particle size distribution curves of the crusher products are shown in Figure 5.

Table 5 summarizes the content of irregular grains in different fractions of products obtained from the impact crusher. The content of non-formed grains was measured using slotted sieves in accordance with the PN-EN 933-3:2012 [33] standard (flakiness index FI), with grains below 4 mm (outside the scope of the standard) included in the analysis for the purposes of the project. The formed irregular grains in the crushing process depend mainly on the physical and mechanical properties of the raw material (hardness, flakiness, toughness, structure, texture), but also on the type of crusher and its technical and technological parameters [34]. Therefore, it can be seen (Table 5) that the highest proportion

of irregular grains (from 15–25%) was observed in the finest fractions of 2–8 mm. This is a known phenomenon that aggregate producers have problems with, as the fine commercial grades are the most difficult to meet the standard requirements. The objective of this project was to reduce the content of unshaped grains to at least 3% in the Formator installation. The proportion of irregular grains also depends on the comminution degree of the raw material. It was observed that the highest content of irregular grains in the 2–8 mm class (FI = 25% sandstone, FI = 24% dolomite) was obtained for high comminution degree S50 values 6.7 and 4.5, and the lowest values FI = 15% (gravel), FI = 17% (granite) for the lowest comminution degree S50 values 3.3 and 3.9. Similar trends are found for the wider grain fraction 2–16 mm where the highest FI ratios are found for dolomite 17.8%, sandstone 16.2%, and limestone 15.2%. Here, the raw materials were crushed at the highest values of comminution degrees (Tables 4 and 5).

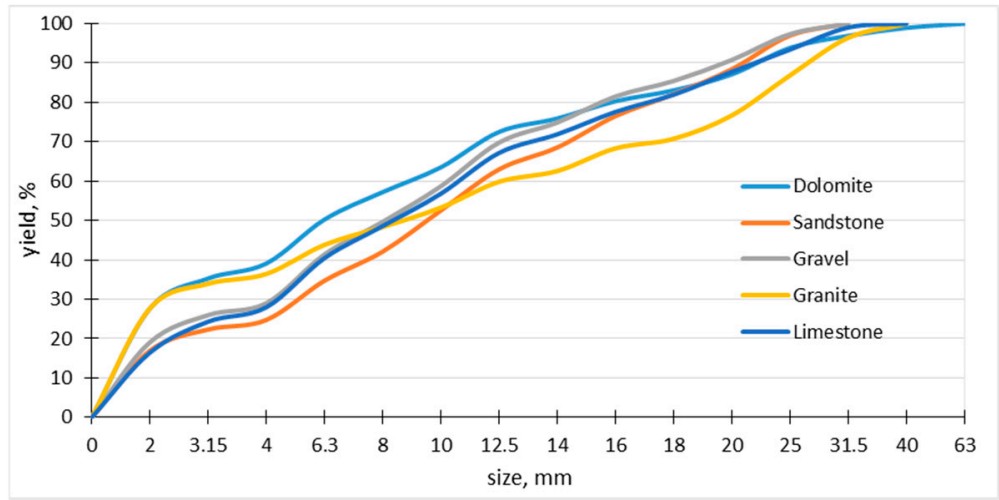

**Figure 5.** Particle size distribution curves of five comminution products in an impact crusher directed to the WSR screen.

**Table 5.** Contents of irregular grains in individual fractions obtained in an impact crusher.

| Crusher Product [mm] | Flakiness Index FI [%] | | | | |
|---|---|---|---|---|---|
| | **Dolomite** | **Sandstone** | **Gravel** | **Granite** | **Limestone** |
| 2–8 | 24.0 | 25.0 | 15.3 | 17.0 | 18.9 |
| 8–16 | 9.8 | 9.7 | 10.8 | 10.3 | 11.1 |
| 2–16 | 17.8 | 16.2 | 13.0 | 13.7 | 15.2 |
| 2–31.5 | 14.9 | 13.5 | 13.6 | 15.7 | 13.7 |

The products obtained after crushing in the impact crusher were transported by a belt conveyor to the WSR screen, where they were sieved into fractions, 0–8, 8–12, 12–16, and +16 mm. The 8–12 and 12–16 mm fractions were sifted on slot screens (longitudinal rectangular mesh), from which regular and irregular grains are separated. These fractions are then combined with respect to shape into fractions of 8–16 mm RP (with regular grains) and 8–16 mm IP (with non-formed grains). Figures 6 and 7 present selected photographs of the product received. Figure 8 presents examples of the samples for analysis.

All of the obtained grain composition curves for the 8–16 mm sieving products of the upper-screening (RP) and bottom-screening (IP) are summarized in a graph (Figure 9). From the analyses, it can be concluded that granite with irregular grains had the finest grain size and dolomite with regular grains had the coarsest grain size. The 8–16 mm IP granite had the highest content of sub-grain up to 22%, which was not completely screened (outcrop of fractions below 8 mm). This is related to the higher throughput (about 100 t/h, Table 3). Comparing the remaining diagrams with each other, it should be stated that all

of the passing material, i.e., those containing irregular grains, are considerably finer than the retained material on the screen surface, i.e., regular ones, as they contain about 10% of undersize. On the other hand, the retained material with regular grains contains from about 1 to 5% of undersize, which indicates the high quality of the aggregates. The content of 5% of sub-grain was recorded for limestone and granite with the screening capacity of about 100 t/h.

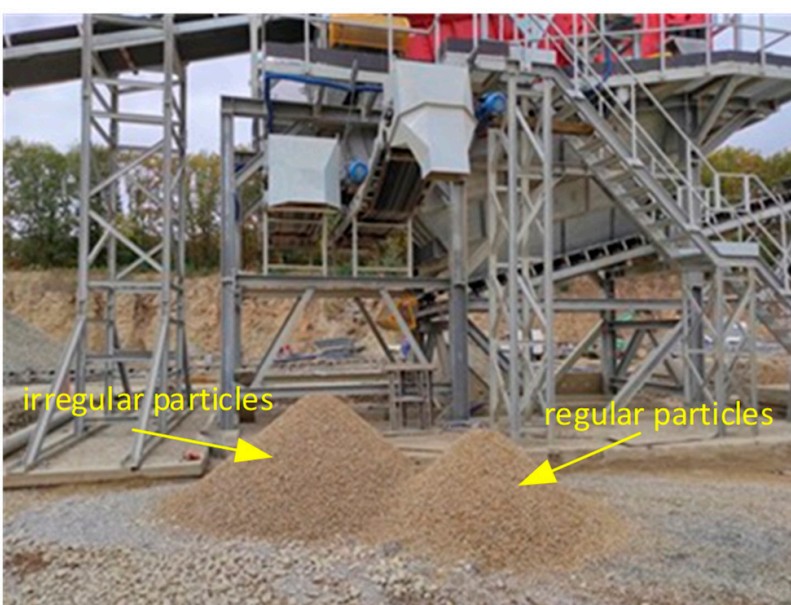

**Figure 6.** Limestone tests 8–16 mm regular and irregular particles.

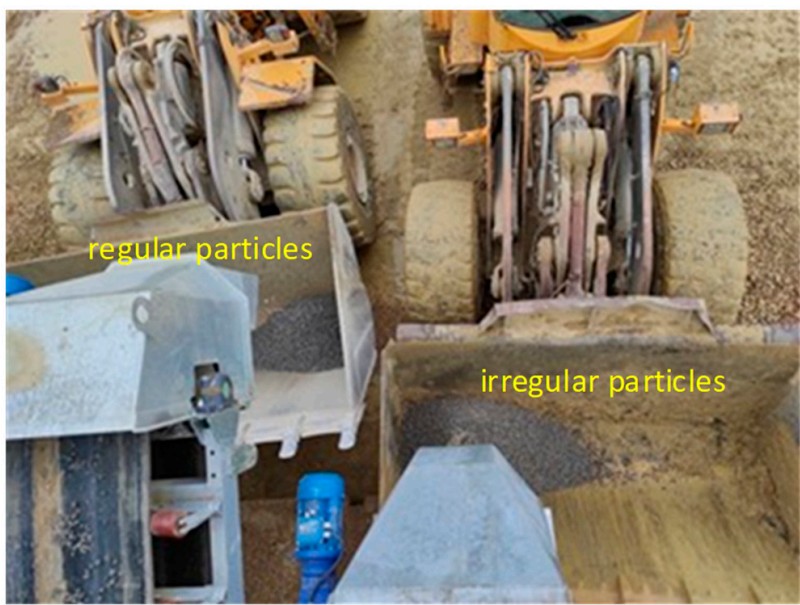

**Figure 7.** Sandstone tests 8–16 mm regular and irregular particles.

Table 5 presents the content of irregular grains for the five final products (FI flakiness index). The final products are the retained material of the WSR 8–16 mm RP screen, on which regular grains retain. All of these products have flakiness indexes of less than 1.5%, the intended goal has been achieved and this means that the content of regular grains in these products exceeds 98%. The lowest contents were obtained for granite (0.2%), limestone (0.3%), and sandstone (0.5%). For gravel, the highest value was 1.3%. The content

of irregular grains for the product including under-screen (6.3–16 mm) was also calculated and ranges from 8 to 17%. Moreover, Table 6 summarizes the content of irregular grains that occurs in the screening product of the screen and comes from the process of sifting these grains on a given screening deck (irregular grains screened on the screen fall through its openings and accumulate in the screening product). In accordance with the idea of the invention, these products can be crushed again or used as aggregates. The values of the proportion of irregular grains are not high, as they range from 9% for granite to 17% for gravel. It can be concluded that these values have typical aggregates produced in processing plants [13,14,28]. The partial shares (6.3–16 mm) reduce the content of irregular grains by 1%, with the exception of limestone (increase by 0.4%).

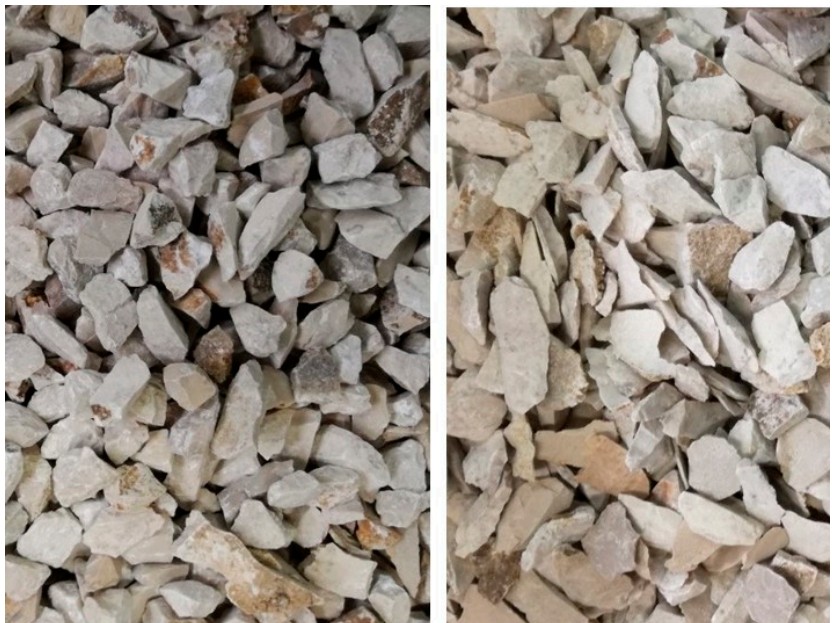

**Figure 8.** Final product: Limestone 8–16 mm regular (**left**), irregular (**right**).

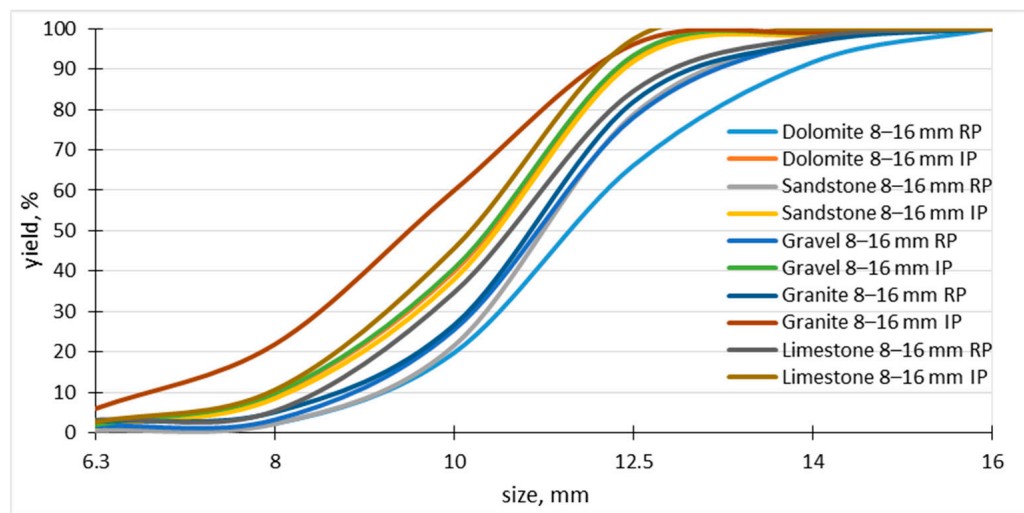

**Figure 9.** Particle size distribution curves of five screening products in the fraction 8–16 mm (WSR screen).

It is worth noting that the content of irregular grains in typical processing plants reaches about a dozen or more percent in coarse products, and in fine grains they often exceed 20%. For comparison, Table 7 summarizes the content of irregular grains for typical

products obtained before refinement in the WSR specialized screen, i.e., these aggregates would be obtained if a typical vibrating screen was used in the technological system. The content of irregular grains would range from 10–11%. In addition, Table 8 summarizes the flow balance of the 8–16 mm product (mass yield) with separation into fractions with regular particles (RP) and irregular particles (IP).

**Table 6.** The flakiness index FI for the five final products of 8–16 mm of the *Formator* technological system.

| Product [mm] | Irregular Particles Content [%] | | | | |
|---|---|---|---|---|---|
| | Dolomite | Sandstone | Gravel | Granite | Limestone |
| 8–16 RP upper-screen product | 1.0 | 0.5 | 1.3 | 0.2 | 0.3 |
| 8–16 RP upper-screen product with undersize grains (6.3–8) | 1.0 | 0.5 | 1.3 | 0.2 | 0.3 |
| 8–16 IP bottom-screen product | 10.3 | 16.1 | 17.4 | 9.1 | 11.8 |
| 8–16 IP bottom-screen product with undersize grains (6.3–8) | 9.8 | 15.2 | 16.5 | 8.2 | 12.2 |

**Table 7.** The flakiness index (FL) values of products received from conventional crushing and screening of plants compared with those received from the innovative one.

| Product [mm] | Irregular Particles Content [%] | | | | |
|---|---|---|---|---|---|
| | Dolomite | Sandstone | Gravel | Granite | Limestone |
| 8–16 RP innovative | 1.0 | 0.5 | 1.3 | 0.2 | 0.3 |
| 8–16 typical | 9.8 | 9.7 | 10.8 | 10.3 | 11.1 |

**Table 8.** Mass balance with separation into fractions with regular particles (RP) and irregular particles (IP).

| Product [mm] | Mass Yield of RP and IP in Products [t/h] | | | | |
|---|---|---|---|---|---|
| | Dolomite | Sandstone | Gravel | Granite | Limestone |
| 8–16 RP | 16.41 | 22.93 | 15.53 | 18.14 | 25.49 |
| 8–16 IP | 2.09 | 4.57 | 3.57 | 1.86 | 3.51 |
| 8–16 RP + IP | 18.5 | 27.5 | 19.1 | 20.0 | 29.0 |

*3.2. Optimization of Required Machine Parameters*

As a result of the tests, the necessary operating parameters of the machines were optimized for their proper functioning. The most important issues included the solution of the problem of blocking of the mesh screens with difficult grains during the process of screening regular and irregular grains in the WSR screen. Difficult grains are grains of similar size to a given screen opening, which cause clogging of the sieves and a decrease of the screening efficiency. Due to the specificity of the technological process, each narrow fraction that is matched to the appropriate width of the slot contains 100% of grains difficult to sift. Therefore, the first screening tests were characterized by too strong blocking of the mesh with difficult grains, as shown in the photographs (Figure 10).

Factors affecting the efficiency of the screening process include [35–37]:

- Dynamic parameters of the screen deck;
- Layer thickness of the screened material;
- Type of screen deck;
- Properties of the material (such as surface moisture, hardness and compaction, grain shape and range of grain size) to be screened.

Since the last three factors cannot be changed due to the nature of the process, the solution had to be found in the dynamics of the screen. In vibrating screens, the movement of grain on the screen is caused by inertia forces, which are the result of periodic movement

of the screen. A value of vibration amplitude that is significantly low influences the lowering of screening process effectiveness by blocking of sieves with grains at a given output or lowering of screening output at assumed screening effectiveness. In screen machines, the transport of the material on the surface of the screen is usually achieved by means of vibrators causing harmonic vibrations or shafts with unbalanced masses, which are directed at an angle to the surface of the screen, while the screen can also be inclined to the horizontal. To parameterize the operation of the screen the dynamic index is determined, which is the ratio of the maximum acceleration of the screen to the acceleration of the earth. The dynamic index of the screen also informs the values of load of the screen construction by inertia forces. Depending on the type of screen motion, we use the feed rate ($u_1$) (rectilinear screen motion) and toss index ($u_2$) (circular, elliptical screen motion), which is the ratio of maximum screen acceleration to the acceleration due to gravity.

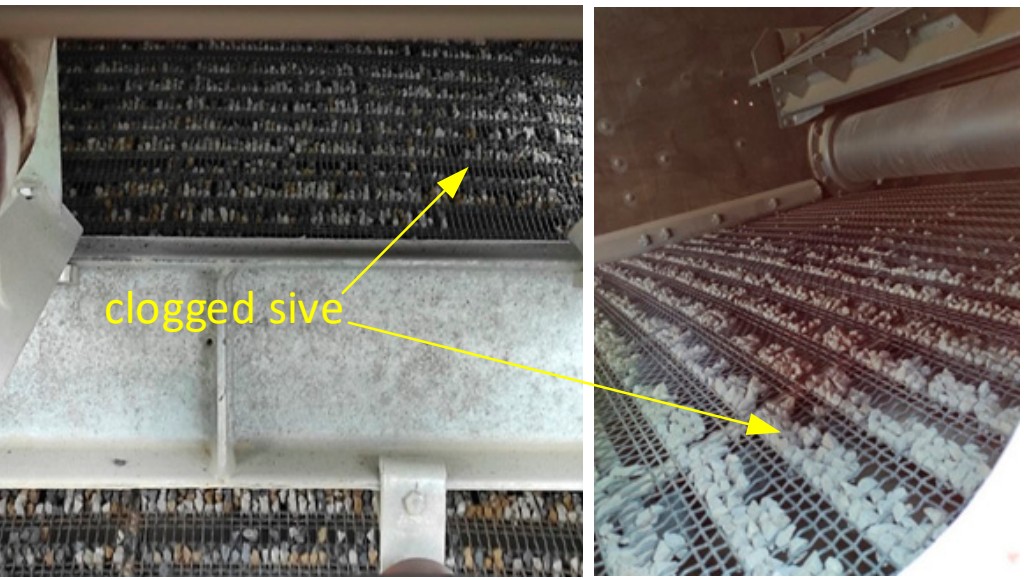

**Figure 10.** Difficult grains blocking slot screen decks (toss index 4.4).

For screens with circular vibration trajectory the toss ratio $u_2$ has the following form [35]:

$$u_2 = \frac{r\omega^2}{g\cos\beta} > 1 \tag{1}$$

where $r$ is the radius of vibration, mm; $\omega$ is the angular velocity, rad/s; $g$ is acceleration due to gravity, m/s$^2$; and $\beta$ is the angle of inclination of the sieve to the horizontal.

The pitch of the riddle is affected by unbalanced masses, which change the radius of its oscillation. Therefore, this radius can be adjusted by adding or removing masses on the eccentric shaft. Since the radius of vibration measured on the riddle was 4 mm, which provided a toss ratio of more than 4.4, it was recommended to gradually increase the unbalanced masses on the shaft and study the cleaning of the sieves (Figure 11). The data are summarized in Table 9 along with the calculated toss ratio $u_2$.

**Table 9.** Operating parameters of WSR screen.

| Radius of Vibration $r$, [mm] | Angular Velocity $\omega$, [rad/s] | Angle of Inclination of the Sieve to the Horizontal, $\beta$ [°] | Toss Index $u_2$ |
|---|---|---|---|
| 4 | 102.6 | 15 | 4.4 |
| 5 | 102.6 | 15 | 5.5 |
| 6 | 102.6 | 15 | 6.6 |

The best effect of sieve cleaning of "difficult" grains was obtained for a vibration radius of 6 mm, which raised the toss ratio to a value of 6.6 acceleration due to gravity. A photo of

the conducted tests and the sieve cleaning effects are illustrated by the photographs shown in Figure 12.

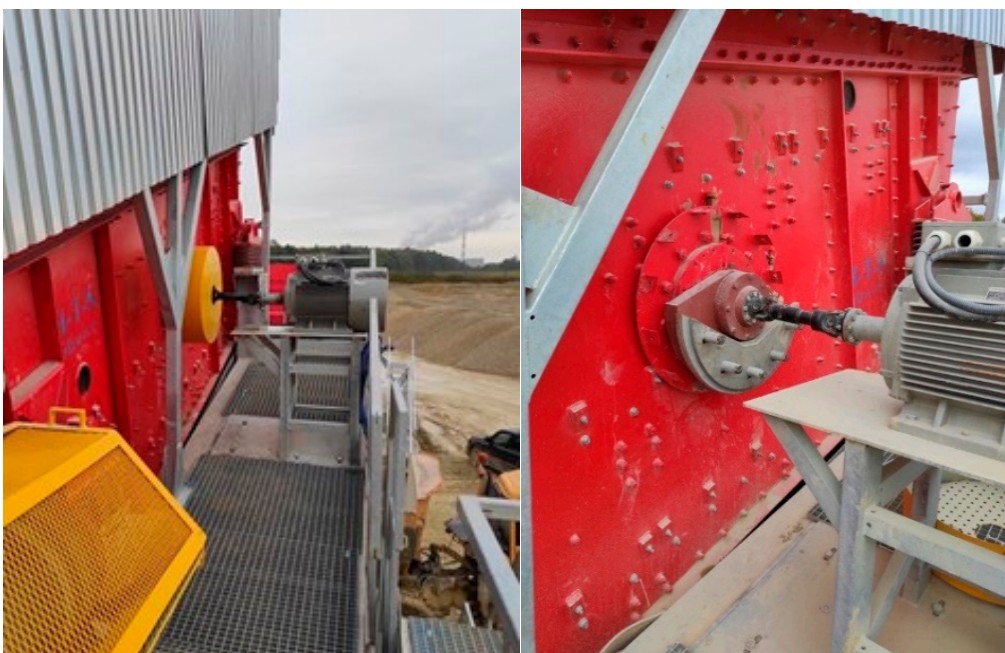

**Figure 11.** Change of unbalanced masses on the eccentric shaft of the screen.

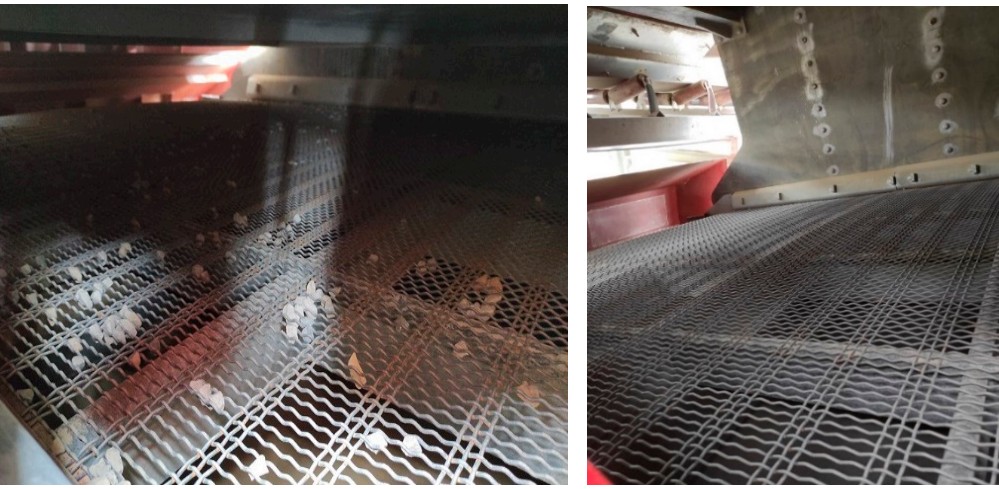

**Figure 12.** Photographs showing the condition of sieve blockage with difficult grains: At a toss index of 5.5 (**left**), at a toss index of 6.6 (**right**).

## 4. Conclusions

- The installation produced at the Dolomite Mine in Imielin has innovative solutions in terms of products and processes, in accordance with the Oslo handbook [38]. It is a novelty that has not been used to date in the world due to its unique features and functionality as compared with the solutions available on the domestic (Polish) and foreign markets.
- By adjusting the appropriate dimensions of the screen (or screen decks), the capacity of the technological crushing and screening system can be increased by at least 30%. The increase in capacity depends on the content of regular grains, which can be taken out of the system as a final product, since there is no need to further crush them in other crushers.

- Tests conducted on the crushing and production of molded aggregates in the innovative WSR screen have shown that aggregates produced from five different rock materials have very low FI flakiness indexes.
- The final products are the upper-screen products of the 8–16 mm RP deposits, on which the regular grains remain. All of these products have flakiness indexes of less than 1.5% (Table 6), this indicates that the content of regular grains in these products exceeds 98%.
- The assumption for the construction of this system was the possibility of producing aggregates with a content of non-formed grains below 3%. The lowest contents were obtained for granite (0.2%), limestone (0.3%), and sandstone (0.5%). The highest value of 1.3% was obtained for gravel.
- It should be highlighted that the obtained low values of irregular grains practically do not occur in any mineral processing plants, thus they can be considered as innovative high-cubic aggregates.

## 5. Patents

The following patent granted in Poland is related to this paper: Author: Gawenda T.: Wibracyjny przesiewacz wielopokładowy, AGH w Krakowie. Patent No. PL 231748 B1 granted on 12 June 2018.

**Author Contributions:** Conceptualization, T.G.; methodology, T.G., A.S. (Agata Stempkowska) and D.S.; validation, D.F. and A.K.; formal analysis, T.G., D.S. and D.F.; investigation, T.G., A.S. (Agata Stempkowska), D.F., A.K. and A.S. (Agnieszka Surowiak); data curation, D.S. and D.F.; writing—original draft preparation, T.G. and A.S. (Agata Stempkowska); writing—review and editing, T.G. and D.S. All authors have read and agreed to the published version of the manuscript.

**Funding:** The paper is the effect of completing the NCBiR Project, contest no. 1 within the subaction 4.1.4 "Appliaction projects" POIR in 2017, entitled "Elaboration and construction of the set of prototype technological devices to construct an innovative technological system for aggregate beneficiation along with tests conducted in conditions similar to real ones". The Project is co-financed by the European Union from sources of the European Fund of Regional Development within the Action 4.1 of the Operation Program Intelligent Development 2014–2020.

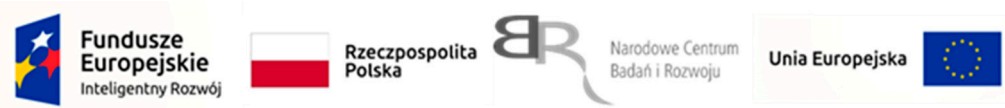

**Data Availability Statement:** The data presented in this study are available on request from the corresponding author.

**Conflicts of Interest:** The authors declare no conflict of interest.

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
