# Peer review of "Assessment of Operational Effectiveness of Innovative Circuit for Production of Crushed Regular Aggregates in Particle Size Fraction 8–16 mm"

_minerals, doi:10.3390/min12050634_

Round 1
Reviewer 1 Report
I have made several comments to your article. Please see the recommendations, which I have highlighted on the submitted file, since I consider that your work needs extensive reconsideration.

Author Response
Dear Reviewer 1
Thank you for your involvement in reviewing our publication, please find attached our answers to questions and comments
Authors

Reviewer 2 Report
The paper reports a set of laboratory experiments for the study of “Assessment of operational effectiveness of innovative circuit for production of crushed regular aggregates in particle size fraction 8-16 mm”
The shape of the particles conditions the properties of the aggregates. The authors study the effect of the rock type, the crushing rate, and the use of special screen in the particles shape.
This article is interesting, it can be accepted with minor changes.
I have listed some comments below.
Page 8, Line 196
“This is related to the higher efficiency (about 100 t/h)”
Clarify this statement.
Page 8 Line 209
All of these products have flakiness indexes of less than 1.5 % (bold in the table 5),
Bold letter is not visible.
Figure 6 and 7
The shape of the particles is not visible. Figures 6 an d7 are dispensable.
For limestone you could present a figure similar to that of sandstone (Figure 8).
Throughout the work, there is no mass balance.
For the 8-16 mm fraction it produces regular particles (RP) and irregular particles (IP). What is the mass of regular material and irregular material?
In Figure 1 and Figure 2 present the light contaminant separator (SEL) and the separator of hard-to-separate fractions (SET). But don't talk about those separators again, how do they work? what's their objective, results?
Figure 2
Is the 2-4 mm IP fraction not produced?
Is only the 4-8 mm IP fraction produced?
Author Response
Dear Reviewer 2
Thank you for your involvement in reviewing our publication, please find attached our answers to questions and comments
Authors

Reviewer 3 Report
This paper analyzed a modern and unique technological system producing common aggregates at the Imielin Dolomite Mine. It was carried out on the crusher and screen on the example of production of aggregates with grain size 8–16 mm from dolomite, granite, limestone, sandstone and gravel. However, the overall scientific content and degree of novelty are not high enough, and the results were poorly presented. After a careful examination, I find that this manuscript can not be accepted for publication.
Author Response
Dear Reviewer 3
Thank you for your involvement in reviewing our publication, please find attached our comments
Authors

Reviewer 4 Report
It is a nice paper on practical application and innovation. There a few things, I would like to suggest authors to include in the revised manuscript.
- In introduction: please provide a short tabular comparison between your innovate machine/equipment and other commercially available machine with their merits and demerits.
- Introduction Line 42-43: please provide a reference or any substantial data to demonstrate "In general, 42 the more compact the raw material, the more difficult it is to obtain cubic aggregate from it"
- Figure numbers are highlighted with color - need to remove the highlights.
- Can you discuss the limitations of your equipment/circuits on a paragraph?
Author Response
Dear Reviewer 4
Thank you for your involvement in reviewing our publication, please find attached our answers to questions and comments

Round 2
Reviewer 1 Report
Ok, acceptable, to a significant extent, your adaptation to my suggestions.
Reviewer 3 Report
The manuscript's quality has been substantially improved. I recommend its acceptance for publication in its present form.